# Dissipation Dynamics and Dietary Risk Assessment of Kresoxim-Methyl Residue in Rice

**DOI:** 10.3390/molecules24040692

**Published:** 2019-02-15

**Authors:** MingNa Sun, Lu Yu, Zhou Tong, Xu Dong, Yue Chu, Mei Wang, TongChun Gao, JinSheng Duan

**Affiliations:** 1Institute of Plant Protection and Agro-Product Safety, Anhui Academy of Agricultural Sciences, Hefei 230031, China; sunmingna@126.com (M.S.); yulu718@126.com (L.Y.); tongzhou0520@163.com (Z.T.); dongxu929@163.com (X.D.); chuychu@163.com (Y.C.); wangmeinc@sina.com (M.W.); 2Key Laboratory of Agro-Product Safety Risk Evaluation (Hefei), Ministry of Agriculture, Hefei 230031, China

**Keywords:** rice, kresoxim-methyl, dissipation, risk assessment

## Abstract

Kresoxim-methyl is a high-efficiency and broad-spectrum fungicide used for the control of rice fungal diseases; however, its residues after application potentially threaten human health. Investigations on the dissipation of kresoxim-methyl residue in rice field systems and dietary risk assessment of kresoxim-methyl in humans are limited. The present study employed the QuEChERS-GC-MS/MS method for residue analysis of kresoxim-methyl in rice plants, brown rice, and rice husks. The samples were extracted with acetonitrile and purified by PSA, C_18_ column, and GCB. The average recovery of the spiked target compounds in the three matrices was between 80.5% and 99.3%, and the RSD was between 2.1% and 7.1%. The accuracy and precision of the method is in accordance with the requirements of residue analysis methods. Dissipation dynamic testing of kresoxim-methyl in rice plants indicated a half-life within the range of 1.8–6.0 days, and a rapid dissipation rate was detected. Dietary intake risk assessment showed that the national estimated daily intake (*NEDI*) of kresoxim-methyl in various Chinese subpopulations was 0.022–0.054 μg/(kg bw·days), and the risk quotient (*RQ*) was 0.0000055–0.00014%. These findings indicate that the risk for chronic dietary intake of kresoxim-methyl in brown rice is relatively low. The present study provides information and theoretical basis for guiding the scientific use of kresoxim-methyl in rice fields and evaluating its dietary risk in brown rice.

## 1. Introduction

Rice (*Oryza sativa*) belongs to the herbaceous rice genus and is the most important rice food crop. Rice is one of the three major food crops in China, and its planting area accounts for 27% of the country’s total grain planting area [1]. However, the occurrence of plant diseases (e.g., sheath blight and rice blast) has led to a sharp decline in the yield and quality of rice, thereby posing a major threat to food security and safety. Chemical control is currently considered as the most effective way to prevent and control rice diseases, there were some chemicals, including tebuconazole and azoxystrobin [2], have been applied and detected in rice bio-system.

Kresoxim-methyl is a methoxy acrylate fungicide developed by BASF (Stuttgart, Germany) and has been shown to have highly efficient and broad-spectrum antifungal activity. The EC50 values for kresoxim-methyl in inhibiting mycelial growth of the *M. oryzae* isolates was 0.024–0.287 μg mL^−1^ [3]. It inhibits fungal pathogens such as oomycetes, and ascomycetes by regulating their mitochondrial respiration [4,5] and is widely used in vegetables, fruits and grains [6]. In China, kresoxim-methyl has been registered and used for the control of rice sheath blight [7], although current understanding of the potential risk of its resides in rice to human health after application is limited. Therefore, it is particularly important to study the residual changes involving kresoxim-methyl in the rice field system after the application and to assess its residual dietary risk.

Extensive studies on the residues of kresoxim-methyl in plant-derived agricultural products have been conducted to date. GC-ECD (Gas Chromatography-Electron Capture Detector) has been used to analyze the residues of kresoxim-methyl in plum [8] and cotton seed [9], and HPLC-UV (High-performance liquid chromatography-Ultra Violet) has been employed to analyze the residue of kresoxim-methyl in apple juice [10] and soil [11]. Mass spectrometry has also been conducted for the detection of kresoxim-methyl. LC-MS/MS (Liquid chromatography coupled with tandem mass spectrometry) has been utilized in the analysis of residues of kresoxim-methyl in grapes [12]. Meanwhile, the sample procedure including QuEChERS, DLLME (dispersive liquid–liquid microextraction), EADLLME (effervescence-assisted dispersive liquid–liquid microextraction) have been used in previous reports. The dissipation dynamics of kresoxim-methyl residue in apples [13], strawberries [14], grapes [15] and tomatoes [16]. However, residue analysis of kresoxim-methyl in the rice field system and dietary risk assessment in rice have not been conducted to date. With the development of determination, some advanced technologies including the QuEChERS [17] and GC-MS/MS [18] (Gas chromatography coupled with tandem mass spectrometry), have been applied extensively. This study builds an efficient and sensitive method for determination of kresoxim-methyl, and provides a basis for the rational use of kresoxim-methyl on rice and its dietary safety.

## 2. Results and Discussion

### 2.1. Method Optimization

The purification adsorbents commonly used to date in the QuEChERS method include PSA (Primary secondary amine-bonded silica), C18, and GCB (Graphitized carbon black). PSA is used to remove polar interferences such as sugars, organic acids, and fatty acids; C18 is used to remove non-polar organic compounds such as fats and lipids; GCB is used to remove pigment compounds such as chlorophyll and carotenoids.

In this study, the purification effects of PSA, C18, PSA + C18, and PSA + C18 + GCB on the complex substrates (rice plants, brown rice, and rice husks) were studied here when 0.1 mg/kg of kresoxim-methyl was added into the substrates. The average recovery of the target compounds and the matrix effects in different treatment groups are shown in Figure 1. When PSA is used alone, the matrix effect is larger than other treatments, and the recovery rate of kresoxim-methyl is higher (108.2–129.5%). The matrix effect of brown rice and rice husk is obviously reduced after C18 purification, but there is also some certain absorption of kresoxim-methyl in both groups. The recovery rates of kresoxim-methyl in the groups of brown rice and rice husk were less than 75%, and thus the amount of C18 was reduced for purification. When PSA and C18 were combined to use, the recovery rate and purification effect of kresoxim-methyl in the groups were higher than those after PSA or C18 treatment. However, the recovery rate in rice plants group remained high, and no pigment adsorption could be detected. When the PSA + C18 + GCB purification method was used, the matrix effect was minimal, it illustrated that the PSA + C18 + GCB adsorbent has a better purification effect. The average recovery rates of kresoxim-methyl in the samples of rice plants, brown rice, and rice hull were within the range of 90.7–96.1%, which met the requirements for testing pesticide residue. Based on these results, PSA + C18 + GCB (ratio was 5:5:1, weight was 110 mg) was used as adsorbent for sample purification in this study.

### 2.2. Method Validationlimit of Quantification

The linear relationship and limit of quantification (LOQ) of kresoxim-methyl in the three matrices are shown in Table 1. Kresoxim-methyl showed a good linear relationship within the range of 0.002–0.5 mg/L, and *r^2^* was 0.9996–0.9998. The limit of quantification of kresoxim-methyl in the three matrices was 0.005 mg/kg.

### 2.3. Accuracy and Precision

The results of the spike-and-recovery test of kresoxim-methyl are shown in Table 1. The average recovery of spiked kresoxim-methyl in rice plants, rice hulls, and brown rice was 84.1–99.3%, 85.8–91.7%, 80.5–89.0%, respectively. Their relative standard deviation in the groups of rice plants, rice hulls, and brown rice were 2.72–4.21%, 2.12–4.21%, and 5.43–7.11%, respectively. The accuracy and precision of the method met the requirements for pesticide residue analysis [19]. A typical chromatogram of the spike-and-recovery test is shown in Figure 2.

### 2.4. Digestion Dynamics in Rice Plants

According to JMPR, the definition of the residue in plant origin is only maternal kresoxim-methyl, then we could not consider the metabolites in the degradation of kresoxim-methyl in rice [20]. We set three quality control samples with the concentration of 100 ng/g during the process of determination. The residual amount of kresoxim-methyl on rice plants decreased with time, and the degradation process accorded with the first-order reaction kinetics equation (The correlation coefficient ranged from 0.8512–0.9480). The results are shown in Figure 3. The original deposition of kresoxim-methyl in rice plants was 6.511–20.096 mg/kg. After 21 days of application, the residual dissipation rate was over 90%, and the half-life of kresoxim-methyl was 1.8–6.0 days. The results tested in different years showed that the dissipation trend of the three test points is basically the same in the same year. In 2016, the dissipation rate of kresoxim-methyl was slightly faster than that of 2017, which may be related to the weather conditions (such as temperature, humidity, wind speed, rainfall) of the year.

### 2.5. Final Residue Test

The results of final residue test showed that the residue of kresoxim-methyl was less than 0.005–0.226 mg/kg in the rice plant, less than 0.005–0.008 mg/kg in brown rice, and less than 0.005–0.460 mg/kg in the rice husk after 14 days from the last application. The residual amount of kresoxim-methyl is less than 0.005–0.124 mg/kg in the plant, less than 0.005–0.006 mg/kg in brown rice, and less than 0.005–0.0455 mg/kg in rice husk at 21-days after application. The residual amount of kresoxim-methyl is less than 0.005–0.096 mg/kg in plants, less than 0.005 mg/kg in brown rice, and less than 0.01–0.237 mg/kg in rice husks at 28-days after application. The residual amount increases with the increase of the application concentration and decreases with the extension of the harvesting interval. The maximum residual amount of kresoxim-methyl is 0.008 mg/kg in brown rice, which is lower than the maximum residue limit of 0.1 mg/kg and 0.01 mg/kg of kresoxim-methyl in brown rice according to China [21] and EU [22], respectively.

### 2.6. Risk Assessment of Chronic Dietary Intake

Kresoxim-methyl is registered to be used on 20 kinds of crops in China such as rice, wheat, strawberry, tomato, and cucumber. Here, the intake amount of kresoxim-methyl derived from rice was evaluated. The standard pesticide residue test results showed that the median residual amount of kresoxim-methyl in the samples collected at 14, 21, and 28 days after application was lower than the lowest limit of quantification (0.005 mg/kg). According to the principle of maximum risk, the STMR value of the dietary risk assessment is 0.005 mg/kg, and the ADI value of kresoxim-methyl is 0.4 mg/kg bw. Combined with dietary patterns in different populations in China, *NEDI* and *RQ* were also calculated. The results in Table 2 showed that the *NEDI* value of kresoxim-methyl in different age groups and genders in China was within the range of 0.022–0.054 μg/(kg bw·days), and *RQ* of kresoxim-methyl was 5.5 × 10^−5^–1.4 × 10^−4^, far less than 1. All the results indicate that the dietary exposure risk of kresoxim-methyl in rice is low and acceptable for humans.

## 3. Materials and Methods

### 3.1. Reagents and Standards

Kresoxim-methyl standard (purity 99.9%, purchased from BASF); C18, PSA (Primary secondary amine), and GCB (Graphitized carbon black) purchased from Agela Technologies (Tianjin, China); anhydrous magnesium sulfate, sodium chloride, acetonitrile, dichloromethane (analytical grade, supplied by Sinopharm Chemical Reagent, China); n-hexane (chromatographical grade, purchased from TEDIA, Woodstock, IL, USA).

Ten micrograms of kresoxim-methyl standard were accurately weighed and dissolved in acetone to obtain a 1000 mg/L standard stock solution, which was kept at 4 °C in the dark. The stability of standard stock solution and the kresoxim-methyl on the matrices were determined every ten days during three months, and the rate of degradation was less than 5%. The standard stock solution of kresoxim-methyl was diluted with the extract solution of rice plant, rice, and rice hull to prepare matrix matching solutions for sample quantification at the various concentrations of 0.002, 0.005, 0.01, 0.02, 0.05, 0.1, 0.2, 0.5 mg/L.

### 3.2. Instrument Conditions

The sample was tested by Shimadzu TQ8040 (Shimadzu, Japan); Rxi-5Sil MS capillary column (30 m × 0.25 mm × 0.25 μm); the initial temperature of the column oven was 60 °C, and the temperature was raised to 150 °C at 30 °C/min. Later the temperature was raised to 250 °C at 10 °C/min for 3 min, and the temperature was further raised to 280 °C at 20 °C/min; the injection port temperature was 270 °C; the carrier gas was helium (purity is 99.999%); the splitless injection was used here, and the injection volume was 1 μL.

Mass spectrometry conditions: electron ionization (EI) mode; transmission temperature was 280 °C; ion source temperature was 230 °C; collision gas was argon (purity 99.999%); solvent delay was 3 min; data acquisition mode was MRM; qualitative ion (*m/z*): 206.1/131.1, 206.1/116.1; quantitative ion (*m/z*): 206.1/131.1.

### 3.3. Field Experiments

The tested 23% kresoxim-methyl suspension was supplied by BASF. Field experiments were conducted in Anhui Province, Hubei Province and Guangdong Province of China in 2016 and 2017. The test site was not less than 30 m^2^ per cell. All the experiments were performed in triplicate. A blank control was set, and a protection line was set between each cell. The application method of kresoxim-methyl suspension was sprayed on stem and leaf using an automatic sprayer during the BBCH code was 33. Kresoxim-methyl suspension at the concentration of 310.5 g a.i./hm^2^ was used in the dissipation test here, which was sprayed once in the rice tillering stage. There was no raining at 2 h after application, and the temperature ranged 23–31 °C. The samples were collected at 2 h, 1 days, 3 days, 5 days, 7 days, 14 days, 21 days, and 30 days after application. The final residue test was set at a low concentration of 207 g a.i./hm^2^ and a high concentration of 310.5 g a.i./hm^2^, which was applied for a total of three or four times. The interval between applications was 7 days. Rice plants, rice and rice hull samples were collected on 14 days, 21 days, and 28 days after the last application. All the samples were preserved at –20 °C.

### 3.4. Sample Processing

Before the pretreatment, plant samples were cut into small pieces for 1cm, then the brown rice and rice husk were broken into powder. Five grams of sample were added into 10 mL of water with 20 mL of acetonitrile. The sample was vortexed for 20 min, and then was extracted by ultrasonic for 10 min with four grams of anhydrous magnesium sulfate. The solution was centrifuged at 4500 rpm for 5 min. The upper extract (2.5 mL) was transferred to a 10-mL centrifuge tube containing 150 mg anhydrous magnesium sulfate, 50 mg PSA, 50 mg C18, and 10 mg GCB. The solution was vortexed for 1 min and then centrifuged at 5000 rpm for 2 min. The supernatant (2 mL) was collected in a glass tube, and blown to dryness. The residue was dissolved in 2 mL of n-hexane. The sample solution was purified by 0.22-μm filter (EMD Millipore, Billerica, MA, USA) under atmospheric pressure, and then it was tested later.

### 3.5. Risk Assessment of Chronic Dietary Exposure

The risk of chronic dietary exposure was calculated according to formulas (1) and (2):(1)NEDI=(∑STMRi×Fi)/bw
(2)RQ=NEDI/ADI

In Equation (1), *NEDI* [μg/(kg bw·days)] is the national estimated daily intake per capita in China; *STMR_i_* (mg/kg) refers to the median value of the standard test residue of the *i*-th grade agricultural products; *F_i_* (g) is the dietary consumption of the *i*-th grade agricultural products for different groups of people in China.

In Equation (2), *RQ* is the risk quotient, and *ADI* is the allowable dietary intake of pesticide per kilogram of body weight, mg/kg bw. When *RQ* ≤ 1, it indicates that the risk is acceptable. The smaller the *RQ* is, the smaller the risk. When *RQ* > 1, it indicates that there is an unacceptable chronic risk. The larger the *RQ*, the greater the risk.

## 4. Conclusions

The QuEChERS method combined with GC-MS/MS technology was used to determine the residual amount of kresoxim-methyl in rice plants, brown rice, and rice husks in this study. The method has the advantages of high sensitivity, accuracy, and precision. The results in the dissipation dynamic test indicated that the dissipation rate of kresoxim-methyl in rice plants was faster than the other matrices, indicating that kresoxim-methyl is an easily degradable pesticide. Based on the residual median value obtained in the final residue test, the chronic dietary intake risk of kresoxim-methyl in rice was evaluated. The results showed that the residue of kresoxim-methyl in brown rice poses a low risk to human health. The evaluation of the environmental toxicity of kresoxim-methyl to non-target organisms will be the direction of future research.

## Figures and Tables

**Figure 1 molecules-24-00692-f001:**
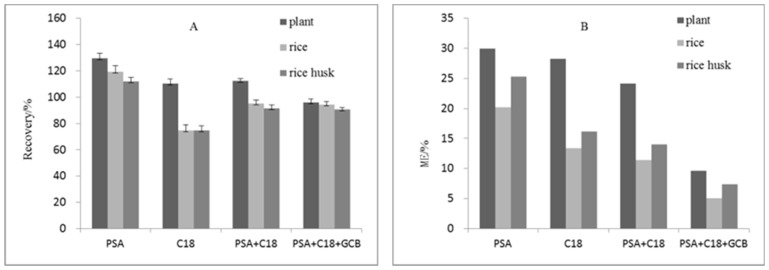
The effects of different purification adsorbents on the recovery of kresoxim-methyl (**A**) and matrix effect (**B**) (*n* = 5). The weight of PSA, C18, PSA + C18 and PSA + C18 + GCB was 50 mg, 50 mg, 100 mg and 110 mg respectively.

**Figure 2 molecules-24-00692-f002:**
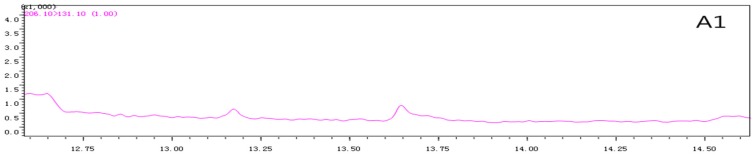
The typical chromatogram of spike-and-recovery test. **A1**, blank rice plants; **A2**, rice plants spiked with 0.005 mg/kg of kresoxim-methyl; **B1**, blank brown rice; **B2**, brown rice with 0.005 mg/kg of kresoxim-methyl; **C1**, blank rice husk; **C2**, rice husk with 0.005 mg/kg of kresoxim-methyl.

**Figure 3 molecules-24-00692-f003:**
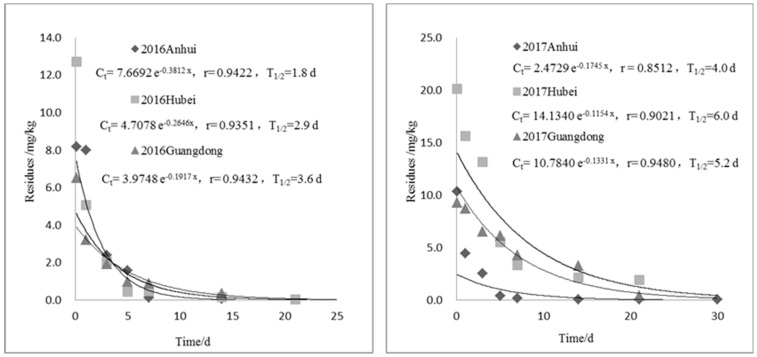
Digestion dynamics of kresoxim-methyl in rice plants.

**Table 1 molecules-24-00692-t001:** Linear range, linear equation, correlation coefficient, recovery, and relative standard deviation (RSD) of kresoxim-methyl.

Matrix	Linear Range (mg/kg)	Regression Equation	Correlation Coefficient (*r^2^*)	Spiked Level (mg/kg)	Average Recovery-Inter (%)	RSD (%)	Average Recovery-Intra (%)	RSD (%)
Plant	0.005–0.5	2323804.0*x* + 5135.8	0.9996	0.005	84.1	4.1	86.2	3.2
				0.05	89.5	2.7	88.2	5.6
				0.5	99.3	3.2	95.2	7.5
Brown rice	0.005–0.5	2559862.7*x* + 5135.8	0.9997	0.005	84.4	7.1	85.2	5.6
				0.05	80.5	6.5	92.3	4.2
				0.5	89.0	5.4	89.3	3.2
Rice husk	0.005–0.5	2291345.6*x* − 4363.7	0.9998	0.005	86.9	2.1	95.6	6.3
				0.05	91.7	3.1	89.5	5.5
				0.5	85.8	4.2	82.3	6.1

**Table 2 molecules-24-00692-t002:** Average intake of brown rice by various populations and estimated exposure and risk quotient of kresoxim-methyl.

Age (y)	Sex	Body Weight (kg)	*F* (g/days)	*NEDI* [μg/(kg bw·days)]	*RQ*
2–3	Male	13.2	135.5	0.051	0.00013
	Female	12.3	133.7	0.054	0.00014
4–6	Male	16.8	179.7	0.053	0.00013
	Female	16.2	159.5	0.049	0.00012
7–10	Male	22.9	230.8	0.050	0.00013
	Female	21.7	212.0	0.049	0.00012
11–13	Male	34.1	266.2	0.039	0.000098
	Female	34.0	238.4	0.035	0.000088
14–17	Male	46.7	308.7	0.033	0.000083
	Female	45.2	240.7	0.027	0.000067
18–29	Male	58.4	309.6	0.027	0.000066
	Female	52.1	260.9	0.025	0.000063
30–44	Male	64.9	316.2	0.024	0.000061
	Female	55.7	278.6	0.025	0.000063
45–59	Male	63.1	314.9	0.025	0.000062
	Female	57.0	272.8	0.024	0.000060
60–69	Male	61.5	274.0	0.022	0.000056
	Female	54.3	242.9	0.022	0.000056
>70	Male	58.5	258.3	0.022	0.000055
	Female	51.0	223.5	0.022	0.000055

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
