# Peer review of "Dissipation Dynamics and Dietary Risk Assessment of Kresoxim-Methyl Residue in Rice"

_molecules, 2019, doi:10.3390/molecules24040692_

Reviewer 1 Report

Molecules

REVIEW

Manuscript ID: molecules-423329

Title: «Dissipation dynamics and dietary risk assessment of kresoxim-methyl residue in rice».

            The article by Sun et al., deals with crucial environmental fate aspects of the pesticide kresoxim-methyl in rice as well as its dietary risk assessment for the consumers. Generally, this work seems to be well planed and organized apart from exhibiting also well linguistic and grammar skills.

            However, the authors should take into consideration some important remarks:

1.    “Introduction”: Since the QuEChERS methodology is described and optimized, some lines could be added in the introduction section concerning the extraction method used.

2.    Sample processing”: The authors should include some information related to the pretreatment of the samples before subjected to the QuEChERS-methodology (e.g. homogenization, lyophilization?) especially since their nature is quite different (plant, brown rice, rice husk).

3.    Sample processing”: How the authors ensured that the samples used for the method optimization and validation were free of the analyte of interest?

4.    Sample processing”: lines 88-89. It does not seem to me that the first step of the methodology (the extraction step) is following the QuEChERS general guidelines. For example, I would like that the authors comment on the reason why they did not add any NaCl along with MgSO4 or any acid as this is a usual procedure in QuEChERS application. Also, the authors should add a reference concerning the unusual procedure of the extraction step they have adopted.

5.    Line 131, paragraph 2.2: The title of this section needs substantial revision and could be rephrased to a more general title such as “method validation”. Otherwise, the correct terminology is “method linearity and limit of quantification”.

6.    I kindly ask that the authors should replace wherever appears in the text the term “limit of quantity” with “limit of quantification”.

7.    Page 4, line 133: The LOQ is not shown in Table 1 as mentioned in the text. Is this the instrument or the method LOQ? Table 1 should be corrected accordingly.

8.    Table 1: Linearity range is referred to the method or the instrument? If it refers to the method linearity it should be expressed in mg/Kg also. If it refers to the instrument is not of crucial importance as to be shown in that table. Instead, it should be replaced by the method linearity. If it is just a printing error, then, it NOT POSSIBLE that the LOQ is higher than the lower border of the linearity. In other words, one cannot quantify under 0,005 mg/Kg, so it not possible to have a linear method response under that value.

9.    Page 4, line 143: the authors should add the reference related to the method validation criteria for the pesticides residues in food.

10.  Page 6, line170: the authors should add the references related to the given maximum residue limit in China, but also for Europe or USA for example.

11.  The authors claim that the detection and quantification of the residues was done by GC-MS/MS. Please add details on how the MS/MS technique was used for the confirmation (if so) of the target pesticide residues. Otherwise, it should be corrected to GC-MS analysis.

      As a whole, this work is recommendable for publication after major revision and only if the above amendments will be taken into consideration by the authors.

Author Response

Dear Editors and Reviewer: Thank you for your letter and for the reviewer’s comments concerning our manuscript entitled “Dissipation dynamics and dietary risk assessment of kresoxim-methyl residue in rice” (ID: molecules-423329). Those comments are all valuable and very helpful for revising and improving our paper, as well as the important guiding significance to our researches. We have studied comments carefully and have made correction accordingly, which we hope meet with approval. The main corrections in the paper and the responds to the reviewer’s comments are listed below point by point: 1) “Introduction”: Since the QuEChERS methodology is described and optimized, some lines could be added in the introduction section concerning the extraction method used.   Answers:Thank you for your useful suggestion. We have stated the application of extraction method in previous studies. 2) “Sample processing”: The authors should include some information related to the pretreatment of the samples before subjected to the QuEChERS-methodology (e.g. homogenization, lyophilization?) especially since their nature is quite different (plant, brown rice, rice husk).. Answers:Thank you for your suggestion. We have added the information of each matrix sample before the pretreatment. 3) “Sample processing”: How the authors ensured that the samples used for the method optimization and validation were free of the analyte of interest? Answers:Thank you for your question. The blank sample was collected in the field which test pesticides were not applied during one year. 4) “Sample processing”: lines 88-89. It does not seem to me that the first step of the methodology (the extraction step) is following the QuEChERS general guidelines. For example, I would like that the authors comment on the reason why they did not add any NaCl along with MgSO4 or any acid as this is a usual procedure in QuEChERS application. Also, the authors should add a reference concerning the unusual procedure of the extraction step they have adopted. Answers:Thank you for your useful suggestion. When we only used the vortex extraction method, the recovery was difficult to reach at 80%, so we added the ultrasonic extraction to improve the efficiency. The anhydrous magnesium sulfate (MgSO4) was applied after the extraction step. 5) Line 131, paragraph 2.2: The title of this section needs substantial revision and could be rephrased to a more general title such as “method validation”. Otherwise, the correct terminology is “method linearity and limit of quantification”. Answers:Thank you for your useful suggestion. We have revised the title of this section according to your suggestion. 6) I kindly ask that the authors should replace wherever appears in the text the term “limit of quantity” with “limit of quantification”. Answers:Thank you for your useful suggestion. We have replaced “limit of quantity” with “limit of quantification” in this paper. 7) Page 4, line 133: The LOQ is not shown in Table 1 as mentioned in the text. Is this the instrument or the method LOQ? Table 1 should be corrected accordingly. Answers:Thank you for your useful question. The LOQ means the method LOQ, and the value is equal with the lowest spiked level (0.005 mg/kg). 8) Table 1: Linearity range is referred to the method or the instrument? If it refers to the method linearity it should be expressed in mg/Kg also. If it refers to the instrument is not of crucial importance as to be shown in that table. Instead, it should be replaced by the method linearity. If it is just a printing error, then, it NOT POSSIBLE that the LOQ is higher than the lower border of the linearity. In other words, one cannot quantify under 0,005 mg/Kg, so it not possible to have a linear method response under that value. Answers:Thank you for your useful question. This is our mistake. The method linearity would listed in Table 1, and the value has been revised to “0.005-0.5 mg/kg”. 9) Page 4, line 143: the authors should add the reference related to the method validation criteria for the pesticides residues in food. Answers:Thank you for your useful suggestion. We have added the reference in this section. 10)  Page 6, line170: the authors should add the references related to the given maximum residue limit in China, but also for Europe or USA for example. Answers:Thank you for your useful suggestion. We have added the reference in this section, and the MRL in Europe. 11) The authors claim that the detection and quantification of the residues was done by GC-MS/MS. Please add details on how the MS/MS technique was used for the confirmation (if so) of the target pesticide residues. Otherwise, it should be corrected to GC-MS analysis. Answers:Thank you for your useful suggestion. The MRM mode was used for detection and quantification using GC-MS/MS, and the technique parameter of the target compound was stated in the section “Instrument conditions”. I hope these revisions will make you accept my article. You can make any revision if you think it is appropriate. Thank you for your help. And I am looking forward to hearing from you. Sincerely, Jinsheng Duan

Reviewer 2 Report

Manuscript Review, Molecules: Dissipation dynamics and dietary risk assessment of kresoxim-methyl residue in rice
l.13: Based on the data presented here “residues after application threaten human health”appears to be an overstatement. The authors should use ‘ residues after application potentially threaten human health. This statement in the Abstract should reflect that which appears at l 40-41 in the manuscript. “the potential risk of its resides in rice to human health after application”.
l.45-46: Where are the abbreviations explained? e.g. GC-ECD
l.55: ditto, PSA, and GCB?
l.60: What was the stability of 1,000 mg/L standard stock solution at 4°C in the dark? How was this determined?
l.72: Argon should not be capitalized.
l.80: What were the specific spraying conditions, including details of sprayer(s) used?
l.81: 310.5 g.a.i/ should be 310.5 g a.i.; gram weight and abbreviation for active ingredient should not be run together.
l.83: These (dose of 207 gai/hm2 and a high dose of 310.5 83 gai/hm2) are not doses, they are concentrations.
l.89: What are the details for the “ultrasonic”? Four grams of anhydrous magnesium sulfate was were added
l.90: What are the details, including source, of the filtration step with a 0.22-μm filter?
l.94: --- blown to dryness
l.95: --- purified by 0.22-μm filter --- Under vacuum or pressure?
l.108: --- commonly used to date in
l.113: --- studied here when ---
l.126: The optimal weight and ratio of PSA+C18+GCB used should be given in the text.
l.129: Figure 1. The effects of different purification adsorbents on the recovery of kresoxim-methyl (A) and matrix effect (B) (n=5). The weight of the adsorbent used should be given. One should be able to understand everything in the figure by the title of the figure; or the title and a footnote to the figure.
l.131: 2.2. Llinear relationship
l.143: “the method met the requirements for pesticide residue analysis”. There should be a reference here to what agency or agencies set these requirements.
l.165: --- at 21-d interval of application? Should this be --- 21-d after application? Also occurs at l.167.
l. 166: 0.0.096? Should this be 0.096 mg/kg?
l.170: --- according to China law. This statement should be referenced,
2
l.190: How much of the dissipation of kresoxim-methyl might be due to its volatility and transport elsewhere as opposed to its degradation? Is anything known about the degradation products (plant metabolites) of kresoxim-methyl and whether or not they retain any pesticide action or residue concerns? These aspects should be addressed in the Discussion.
l.191: There is no statement about the contributions of each author to this study.

Author Response

Dear Editors and Reviewer:

Thank you for your letter and for the reviewer’s comments concerning our manuscript entitled “Dissipation dynamics and dietary risk assessment of kresoxim-methyl residue in rice” (ID: molecules-423329). Those comments are all valuable and very helpful for revising and improving our paper, as well as the important guiding significance to our researches. We have studied comments carefully and have made correction accordingly, which we hope meet with approval. The main corrections in the paper and the responds to the reviewer’s comments are listed below point by point:

1) l.13: Based on the data presented here “residues after application threaten human health” appears to be an overstatement. The authors should use ‘ residues after application potentially threaten human health. This statement in the Abstract should reflect that which appears at l 40-41 in the manuscript. “the potential risk of its resides in rice to human health after application”.  

Answers:Thank you for your useful suggestion. We have revised these sentences according to your suggestion.

2) l.45-46: Where are the abbreviations explained? e.g. GC-ECD

l.55: ditto, PSA, and GCB?

Answers:Thank you for your suggestion. We have explained the abbreviations in this paper.

3) l.60: What was the stability of 1,000 mg/L standard stock solution at 4°C in the dark? How was this determined?

Answers:Thank you for your suggestion. The stability of 1000 mg/L standard stock solution was determined every ten days during three months, and the rate of degradation was less than 5%. We have added this statement in the manuscript.

4) l.72: Argon should not be capitalized.

Answers:Thank you for your useful suggestion. We have revised this mistake according to your suggestion.

5) l.80: What were the specific spraying conditions, including details of sprayer(s) used?

Answers:Thank you for your useful suggestion. We have added the spraying conditions in this section.

6) l.81: 310.5 g.a.i/ should be 310.5 g a.i.; gram weight and abbreviation for active ingredient should not be run together

Answers:Thank you for your useful suggestion. We have revised this mistake in the manuscript.

7) l.83: These (dose of 207 gai/hm2 and a high dose of 310.5 83 gai/hm2) are not doses, they are concentrations.

Answers:Thank you for your useful suggestion. We have revised dosed to concentrations here.

8) l.89: What are the details for the “ultrasonic”? Four grams of anhydrous magnesium sulfate was added

Answers:Thank you for your useful suggestion. We have added the details in the ultrasonic step.

9) l.90: What are the details, including source, of the filtration step with a 0.22-μm filter?

Answers:Thank you for your useful suggestion. We have added the source of a 0.22-μm filter in this sentence.

10)  l.94: --- blown to dryness

Answers:Thank you for your useful suggestion. We have revised this sentence according to your suggestion.

11) l.95: --- purified by 0.22-μm filter --- Under vacuum or pressure?

Answers:Thank you for your useful suggestion. The sample solution was purified by 0.22-μm filter under pressure, and we have added this statement here.

   12) l.108: --- commonly used to date in

Answers:Thank you for your suggestion. But we don’t know how to revise this sentence with your suggestion.

   13) l.113: --- studied here when ---

Answers:Thank you for your suggestion. But we don’t know how to revise this sentence with your suggestion.

   14) l.126: The optimal weight and ratio of PSA+C18+GCB used should be given in the text.

Answers:Thank you for your useful suggestion. We have added the weight and ratio of PSA+C18+GCB in the manuscript.

15) l.129: Figure 1. The effects of different purification adsorbents on the recovery of kresoxim-methyl (A) and matrix effect (B) (n=5). The weight of the adsorbent used should be given. One should be able to understand everything in the figure by the title of the figure; or the title and a footnote to the figure.

Answers:Thank you for your useful suggestion. We have added the information of the adsorbent with a footnote to this figure.

   16) l.131: 2.2. Llinear relationship

Answers:Thank you for your useful suggestion. We have revised the title of section 2.2 to Method validation.

17) l.143: “the method met the requirements for pesticide residue analysis”. There should be a reference here to what agency or agencies set these requirements.

Answers:Thank you for your useful suggestion. We have added a reference here according to your suggestion.

18) l.165: --- at 21-d interval of application? Should this be --- 21-d after application? Also occurs at l.167.

Answers:Thank you for your useful suggestion. We have revised these mistakes in the manuscript.

19) l. 166: 0.0.096? Should this be 0.096 mg/kg?

Answers:Thank you for your useful suggestion. We have revised this mistake here according to your suggestion.

20) l.170: --- according to China law. This statement should be referenced

Answers:Thank you for your useful suggestion. We have added a reference here according to your suggestion.

21) l.190: How much of the dissipation of kresoxim-methyl might be due to its volatility and transport elsewhere as opposed to its degradation? Is anything known about the degradation products (plant metabolites) of kresoxim-methyl and whether or not they retain any pesticide action or residue concerns? These aspects should be addressed in the Discussion

Answers:Thank you for your useful suggestion. According to JMPR, the definition of the residue in plant origin is only maternal kresoxim-methyl, then we could not consider the metabolites in the degradation of kresoxim-methyl in rice.

22) l.191: There is no statement about the contributions of each author to this study.

Answers:Thank you for your useful suggestion. We have added the contributions of each author to this study.

I hope these revisions will make you accept my article. You can make any revision if you think it is appropriate. Thank you for your help. And I am looking forward to hearing from you.

Sincerely,

Jinsheng Duan

Reviewer 3 Report

    Authors presented dissipation dynamics and dietary risk assessment of kresoxim-methyl residue in rice plants, brown rice and rice husks. The manuscript seems to be interesting and correctly written. However, the main concern of the work is design of the experiment which need to be changed and additional experiments are needed. There is  lack of novelty in manuscript.

     The general comment is that paper is limited to a single pesticide residue study in three rice matrices. Author did not perform soil samples analysis in dissipation kinetics studies and did not include metabolites of krezoxim-methyl which are required for risk assessment. Dissipation kinetics studies of pesticide should provide also the information about its degradation in soil and possible metabolites/degradation products which could be more toxic than parent active substance. Krezoxim-methyl has a complex residue definition. The residue definition of kresoxim-methyl is the sum of total kresoxim-methyl and its metabolites, α-[(o-hydroxymethyl)phenoxy]-o-tolyl(methoxyimino) acetic acid (BF 490-2) and α-(p-hydroxy-o-tolyloxy)-o-tolyl(methoxyimino) acetic acid (BF 490-9).

    In the Introduction paragraph Authors stated that the dissipation dynamics and risk assessment of krezoxim-methyl residue was reported only in apples, but was already reported by many researchers e.g. in strawberries in 2018 by Chen et al., grapes in 2014 by Sabale et al., tomatoes in 2016 by Zhu et al. and soil including its metabolites in 2015 by Khandelwal et al.

References

-         Chen XFan XMa YHu J. 2018. Dissipation behaviour, residue distribution and dietary risk assessment of tetraconazole and kresoxim-methyl in greenhouse strawberry via RRLC-QqQ-MS/MS technique. Ecotoxicol Environ Saf. 148:799-804.

-         Sabale Rupali , Shabeer T. P. Ahammed , Utture Sagar C. , Banerjee Kaushik, Jadhav Manjusha R. , Oulkar Dasharath P. , Adsule Pandurang G., Deshmukh Madhukar B. 2014. Dissipation kinetics, safety evaluation, and assessment of pre-harvest interval (PHI) and processing factor for kresoxim methyl residues in grape Environmental Monitoring and Assessment 186,  4, 2369–2374

-         Zhu Xiaodan,  Jia Chunhong,  Duan Lifang,  Zhang Wei,   Yu Pingzhong,  He Min,  Chen Li, Zhao Ercheng 2016. Residue behavior and dietary intake risk assessment of three fungicides in tomatoes (Lycopersicon esculentum Mill.) under greenhouse conditions. Regulatory Toxicology and Pharmacology 81, 284-287

-         Khandelwal Ashish, Gupta Suman , Gajbhiye Vijay T and Gupta Suman2015. Leaching behavior of Kresoxim-Methyl and Acid Metabolite in normal and sludge amended inceptisol soil. IJAEB: 8(1): 1-9.

Other comments:

What are other active compounds authorised for use on rice inChina?

Please, provide some monitoring data of pesticide residue detection in rice.

Please, specify the requirements for testing pesticide residue.

There is lack of quality assurance data of the analytical method.

What stage of rice cultivation was pesticide suspension applied? Please, give BBCH code.

Please, change "matixes" into "matrices" e.g. line 134.

Line 131, point 2.2. The word "linear" should be written in capital letter

In Table 1  in the second column heading is "Linear rang" should be "Linear range"

Author Response

Dear Editors and Reviewer:

Thank you for your letter and for the reviewer’s comments concerning our manuscript entitled “Dissipation dynamics and dietary risk assessment of kresoxim-methyl residue in rice” (ID: molecules-423329). Those comments are all valuable and very helpful for revising and improving our paper, as well as the important guiding significance to our researches. We have studied comments carefully and have made correction accordingly, which we hope meet with approval. The main corrections in the paper and the responds to the reviewer’s comments are listed below point by point:

1)   The general comment is that paper is limited to a single pesticide residue study in three rice matrices. Author did not perform soil samples analysis in dissipation kinetics studies and did not include metabolites of krezoxim-methyl which are required for risk assessment. Dissipation kinetics studies of pesticide should provide also the information about its degradation in soil and possible metabolites/degradation products which could be more toxic than parent active substance. Krezoxim-methyl has a complex residue definition. The residue definition of kresoxim-methyl is the sum of total kresoxim-methyl and its metabolites, α-[(o-hydroxymethyl)phenoxy]-o-tolyl(methoxyimino) acetic acid (BF 490-2) and α-(p-hydroxy-o-tolyloxy)-o-tolyl(methoxyimino) acetic acid (BF 490-9).  

Answers:Thank you for your useful suggestion. According to JMPR (Report 2001), the definition of the residue in plant origin is only maternal kresoxim-methyl, then we could not consider the metabolites in the degradation of kresoxim-methyl in rice. We have added this statement in the discussion. The metabolites you listed was delimited in animal origin.

2) In the Introduction paragraph Authors stated that the dissipation dynamics and risk assessment of krezoxim-methyl residue was reported only in apples, but was already reported by many researchers e.g. in strawberries in 2018 by Chen et al., grapes in 2014 by Sabale et al., tomatoes in 2016 by Zhu et al. and soil including its metabolites in 2015 by Khandelwal et al.

Answers:Thank you for your important suggestion. We have added these reports in the Introduction.

3) What are other active compounds authorised for use on rice in China?

Answers:Thank you for your question. Other active compounds authorised for use on rice in China are such as tebuconazole, azoxystrobin and so on. And we have added this information in the Introduction.

4) Please, provide some monitoring data of pesticide residue detection in rice

Answers:Thank you for your useful suggestion. We have added some monitoring data in the Introduction.

5) There is lack of quality assurance data of the analytical method.

Answers:Thank you for your useful suggestion. We set three quality control samples with the concentration of 100 ng/g during the process of determination. We have added the statement in the manuscript.

6) What stage of rice cultivation was pesticide suspension applied? Please, give BBCH code.

Answers:Thank you for your useful suggestion. We have added the BBCH code in this section.

7) Please, change "matixes" into "matrices" e.g. line 134.

Answers:Thank you for your useful suggestion. We have revised "matixes" into "matrices" according to your suggestion.

8) Line 131, point 2.2. The word "linear" should be written in capital letter

Answers:Thank you for your useful suggestion. We have revised the title to “Method validation”.

9) In Table 1  in the second column heading is "Linear rang" should be "Linear range"

Answers:Thank you for your useful suggestion. We have revised this mistake according to your suggestion.

I hope these revisions will make you accept my article. You can make any revision if you think it is appropriate. Thank you for your help. And I am looking forward to hearing from you.

Sincerely,

Jinsheng Duan

Reviewer 4 Report

The main objective of this work was to assess a QuEChERS-GC-MS/MS method for residue analysis of kresoxim-methyl in rice plants, brown rice and rice husks. Moreover, this study analysed the correct use of kresoxim-methyl in rice fields and its dietary risk in brown rice.

The work is not particularly original. However, the manuscript fits within the scope of the journal and the results can be considered of current interest. Anyway, several problems/doubts should still be solved before obtain a suitable manuscript to be published in Molecules and several major improvements are still needed in order to achieve a suitable manuscript that wake up the interest of a wider scientific community interested in this kind of studies:

-          The term “risk assessment” is wider than those issues evaluated in this work. Therefore, the title of the manuscript is not suitable.

-          The study is not properly contextualised taking into account the analytical work developed, where the sample preparation using QuEChERS and the analytical technique by GC-MS/MS are not properly highlighted. This and other several journals have publications where QuEChERS approach is analysed in detail, being this an issue often neglected (e.g. “Molecules 2018, 23(8), 2009”, etc.), while the GC–MS/MS method is usually used as routine technique for pesticides quantification (e.g. “J. AOAC Int. 98 (2015) 1163-1170”, etc.).

-          The behaviour of pesticides in the environment is highly dependent on the environmental conditions suffered during the study. In this sense, it should be provided data about the temperature, season and rainfall dropped during the trial days, at least as supporting information. These data and intensity of sunlight could be provided from a near meteorological station.

-          It should be provided information about the stability of the kresoxim-methyl on the matrices studied and the number of days that the samples were stored.

-          It would be suitable provided some ecotoxicological data, e.g. IGC50, LC50, LD50 on target organism.

-          What happens with the assessment of the main factors and interactions among the variables, in sample preparation method and chromatographic analysis? They can influence greatly on matrix effect.

-          Graphics of figure 3 are not clear and some regressions are not suitable. Can be provided any explanation to this concern?

-          It should be provided the intra- and inter-day recoveries of the analytical method.

-          A proposal of potential future work attending to the conclusions should be included in the manuscript.

-          The authors should shorten their sentences to be clearer and simpler for the readers.

Conclusion: the work is of interest. However, major revisions are still needed.

Author Response

Dear Editors and Reviewer:

Thank you for your letter and for the reviewer’s comments concerning our manuscript entitled “Dissipation dynamics and dietary risk assessment of kresoxim-methyl residue in rice” (ID: molecules-423329). Those comments are all valuable and very helpful for revising and improving our paper, as well as the important guiding significance to our researches. We have studied comments carefully and have made correction accordingly, which we hope meet with approval. The main corrections in the paper and the responds to the reviewer’s comments are listed below point by point:

1)   The study is not properly contextualised taking into account the analytical work developed, where the sample preparation using QuEChERS and the analytical technique by GC-MS/MS are not properly highlighted. This and other several journals have publications where QuEChERS approach is analysed in detail, being this an issue often neglected (e.g. “Molecules 2018, 23(8), 2009”, etc.), while the GC–MS/MS method is usually used as routine technique for pesticides quantification (e.g. “J. AOAC Int. 98 (2015) 1163-1170”, etc.).  

Answers:Thank you for your useful suggestion. We have added these publications in the Introduction according to your suggestion.

2) The behaviour of pesticides in the environment is highly dependent on the environmental conditions suffered during the study. In this sense, it should be provided data about the temperature, season and rainfall dropped during the trial days, at least as supporting information. These data and intensity of sunlight could be provided from a near meteorological station.

Answers:Thank you for your important suggestion. We have added the information of the environment in this manuscript.

3)  It should be provided information about the stability of the kresoxim-methyl on the matrices studied and the number of days that the samples were stored.

Answers:Thank you for your suggestion. The stability of the kresoxim-methyl on the matrices was determined during three months every ten days, and the rate of degradation was less than 5%.

4) It would be suitable provided some ecotoxicological data, e.g. IGC50, LC50, LD50 on target organism.

Answers:Thank you for your useful suggestion. We have added ecotoxicological data in the Introduction.

5) What happens with the assessment of the main factors and interactions among the variables, in sample preparation method and chromatographic analysis? They can influence greatly on matrix effect.

Answers:Thank you for your useful suggestion. When the PSA+C18+GCB purification method was used, the matrix effect was minimal, it illustrated that the PSA+C18+GCB adsorbent has a better purification effect. We have added this statement in the manuscript.

6) Graphics of figure 3 are not clear and some regressions are not suitable. Can be provided any explanation to this concern?

Answers:Thank you for your useful suggestion. The correlation coefficient ranged from 0.8512-0.9480, and we have described it in the text.

7)  It should be provided the intra- and inter-day recoveries of the analytical method.

Answers:Thank you for your useful suggestion. We have added the intra- and inter-day recoveries of the analytical method in the paper.

8) A proposal of potential future work attending to the conclusions should be included in the manuscript.

Answers:Thank you for your useful suggestion. We have added a proposal of potential future work attending to the conclusions.

9)  The authors should shorten their sentences to be clearer and simpler for the readers.

Answers:Thank you for your useful suggestion. We have revised this mistake according to your suggestion.

I hope these revisions will make you accept my article. You can make any revision if you think it is appropriate. Thank you for your help. And I am looking forward to hearing from you.

Sincerely,

Jinsheng Duan

Round  2

Reviewer 1 Report

Concerning the manuscript submitted to Molecules entitled as "Dissipation dynamics and dietary risk assessment of kresoxim-methyl residue in rice" (Authors: MingNa Sun, Lu Yu, Zhou Tong, Xu Dong, Yue Chu, Mei Wang, TongChun Gao *, JinSheng Duan *), I have the honour to inform you that I believe that the manuscript has been significantly improved and now warrants publication in Molecules. 

Thank you in advance.

Reviewer 4 Report

In my judgment, after review carefully the manuscript, the changes are adequate and the authors have improved it properly. The confusing issues are also clearer. Currently, the article includes all the necessary information for a proper understanding of the work and results are of interest.